# Spatiotemporal continuous estimates of daily 1-km PM$_{2.5}$ from 2000 to present under the Tracking Air Pollution in China (TAP) framework

Qingyang Xiao[1], Guannan Geng[1*], Shigan Liu[2], Jiajun Liu[1], Xia Meng[3], Qiang Zhang[2]

[1]State Key Joint Laboratory of Environment Simulation and Pollution Control, School of Environment, Tsinghua University, Beijing 100084, China

[2]Ministry of Education Key Laboratory for Earth System Modelling, Department of Earth System Science, Tsinghua University, Beijing 100084, China

[3]School of Public Health, Key Laboratory of Public Health Safety of the Ministry of Education and Key Laboratory of Health Technology Assessment of the Ministry of Health, Fudan University, Shanghai 200032, China

*Correspondence to*: Guannan Geng (guannangeng@tsinghua.edu.cn)

**Abstract.** High spatial resolution PM$_{2.5}$ data covering a long time period are urgently needed to support population exposure assessment and refined air quality management. In this study, we provided complete-coverage PM$_{2.5}$ predictions with a 1-km spatial resolution from 2000 to the present under the Tracking Air Pollution in China (TAP, http://tapdata.org.cn/) framework. To support high spatial resolution modelling, we collected PM$_{2.5}$ measurements from both national and local monitoring stations. To correctly reflect the temporal variations in land cover characteristics that affected the local variations in PM$_{2.5}$, we constructed continuous annual geoinformation datasets, including the road maps and ensemble gridded population maps, in China from 2000 to 2021. We also examined various model structures and predictor combinations to balance the computational cost and model performance. The final model fused 10-km TAP PM$_{2.5}$ predictions from our previous work, 1-km satellite aerosol optical depth retrievals, and land use parameters with a random forest model. Our annual model had an out-of-bag $R^2$ ranging between 0.80 and 0.84, and our hindcast model had a by-year cross-validation $R^2$ of 0.76. This open-access 1-km resolution PM$_{2.5}$ data product with complete coverage successfully revealed the local-scale spatial variations in PM$_{2.5}$ and could benefit environmental studies and policy-making.

## 1 Introduction

Air pollution has been a nonnegligible environmental problem around the world. China implemented strict clean air policies in the past decade that considerably improved air quality. To support the policy evaluation and air quality management, we constructed the Tracking Air Pollution in China (TAP) platform (http://tapdata.org.cn/), which provides near real-time air pollutants, i.e., $PM_{2.5}$ and $O_3$, distribution at a 0.1 degree (approximately 10 km) spatial resolution, from the fusion of ground measurements, satellite retrievals, and chemical transport model (CTM) simulations (Geng et al., 2021). The TAP data benefited the evaluations of clean air policies and the characterization of air pollution exposure (Xiao et al., 2020;Xiao et al., 2021b;Xiao et al., 2021c). However, with the improved air pollution control targets that require refined air quality management, the detailed monitoring of air pollution distribution at higher spatial resolutions is urgently needed.

Recent developments in machine learning algorithms and remote sensing techniques have fueled the production of air pollution data at high spatiotemporal resolutions. For example, moderate-resolution imaging spectroradiometer (MODIS) products provide aerosol optical depth (AOD) retrievals at a 3-km resolution, contributing to the prediction of ground-level $PM_{2.5}$ concentrations at the local scale (Xie et al., 2015;He and Huang, 2018;Hu et al., 2019). The multi-angle implementation of atmospheric correction (MAIAC) algorithm provides AOD retrievals at a 1-km resolution and benefits predictions of $PM_{2.5}$ distribution at a 1-km (Wei et al., 2021;Goldberg et al., 2019;Xiao et al., 2017;Bai et al., 2022b) or higher spatial resolution (Huang et al., 2021). Recently, with the Gaofen-5 satellite retrieval, Zhang et al. (2018) predicted the $PM_{2.5}$ concentration at a 160-m resolution. However, most of these high-resolution data products are limited to after 2013 or cover a specific region of China. Few studies have filled the missing predictions resulted from missing satellite retrievals (Bai et al., 2022b;Ma et al., 2022). This discontinuous $PM_{2.5}$ prediction in space and time not only limits the application of $PM_{2.5}$ products in scientific research and environmental management but also biases the characterization of population exposure to $PM_{2.5}$ pollution (Xiao et al., 2017). Additionally, although high-resolution $PM_{2.5}$ prediction models widely included various land use data, e.g., road maps, land cover types, and points of interest (POIs), to describe the local-scale spatial variations in air pollution emissions and air pollution levels, most studies used only one or two years of land use data during the whole study period and ignored the critical variations in them. This lack of temporal variations in land use data may affect the spatial accuracy of high-resolution $PM_{2.5}$ predictions.

In this study, we constructed a high-resolution $PM_{2.5}$ concentration prediction system under the TAP framework in order to provide 1-km resolution full-coverage $PM_{2.5}$ retrievals covering a long time period. To correctly reveal the spatial characteristics of $PM_{2.5}$ distribution at such a high spatial resolution, we processed MAIAC satellite retrievals as well as evaluated and constructed various temporally continuous land-use parameters with statistical and geospatial modelling that have not been included in previous TAP models. By fusing high-resolution MAIAC satellite retrievals, TAP $PM_{2.5}$ products at a 10-km resolution, satellite normalized difference vegetation index (NDVI) products, and various continuous long-term land

use data, we provide 1-km resolution PM$_{2.5}$ predictions from 2000 to the present with complete coverage and timely updates.
The high quality and easy accessibility of our high-resolution PM$_{2.5}$ data could support research on air pollution and
environmental health at local scales and contribute to the management of local air quality.

## 2 Data and Method

The workflow of this study is shown in Fig. 1. First, we processed and assimilated various predictors. The daily scale varying
predictors include satellite retrieval, TAP 10-km PM$_{2.5}$ predictions, and meteorological fields, and the land use variables
include road map, population distribution, artificial impervious area, and vegetation index. In China, the high-speed economic
development in the past several decades has led to significant changes in land use and population distribution. To correctly
reveal the temporal variations in land use parameters and further benefit the description of local-scale PM$_{2.5}$ concentration
variations, we constructed temporally continuous land use predictors through statistical and spatial modeling. We then
optimized the model structure and selected model predictors through various examinations to balance the model performance
and computing cost. With the selected model design, we fitted three models under the TAP framework: the hindcast model
with training data from 2013 to 2020 to predict historical PM$_{2.5}$ concentrations from 2000 to 2014; the annual model with
training data of each corresponding year from 2015 to 2020; and the near real-time model with rolling one-year training data
that provides near real-time PM$_{2.5}$ predictions until the day before present day.

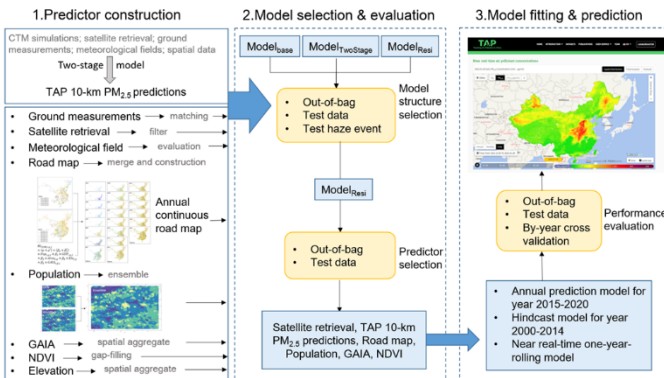


**Figure 1 Workflow of this study.**

## 2.1 Ground measurements of PM$_{2.5}$

The hourly PM$_{2.5}$ measurements from national air quality monitoring stations were downloaded from the China National
Environmental Monitoring Center (http://www.cnemc.cn/). To examine the model's prediction ability in space, we also
collected PM$_{2.5}$ measurements from local air quality monitoring stations operated by local government agencies (Fig. S1). The
hourly PM$_{2.5}$ concentration measurements below 1 µg m$^{-3}$, the lowest measurement limit of most monitors, and above 2000

μg m$^{-3}$ were excluded for quality control of measurements. Identical continuous measurements during at least three hours were also removed. Hourly data generated daily average PM$_{2.5}$ concentrations with fewer than 18 hourly measurements were removed.

In order to examine whether quality of measurements from national monitors and from regional monitors is comparable, we matched the nearest national and regional monitors to compare their measurements. Specifically, we selected such national-regional monitor pairs with distance less than 0.5 degree between them and compared their daily average PM$_{2.5}$ measurements. This comparison illustrates that measurements from regional monitors matched well with measurements from the nearest national monitors with the linear regression coefficient of determination ($R^2$) of 0.89 (slope of 0.99 and intercept of 1.00). The average difference between daily matched regional and national measurements are 1.6 μg m$^{-3}$.

## 2.2 Full-coverage PM$_{2.5}$ predictions at a 10-km resolution

The 10-km resolution PM$_{2.5}$ predictions, which were estimated from a two-stage machine learning modeling system, were downloaded from the TAP website (http://tapdata.org.cn/) (Xiao et al., 2021c). Previous evaluations reported that the PM$_{2.5}$ prediction model with the out-of-bag (OOB) $R^2$ (the $R^2$ of the linear regression between measurements and predictions from trees that did not include these measurements for training) ranged between 0.80 and 0.88 (Geng et al., 2021). These TAP PM$_{2.5}$ data at a 10-km resolution with complete coverage are updated in near real-time.

## 2.3 MAIAC retrievals

The multi-angle implementation of atmospheric correction (MAIAC) (Lyapustin et al., 2011a;Lyapustin et al., 2011b) data were downloaded from the NASA Earthdata site (https://lpdaac.usgs.gov/products/mcd19a2v006/). Only pixels with high-quality retrievals were included (QA. CloudMask=clear and QA. AdjacencyMask=clear) (Kloog et al., 2015;Lyapustin, 2018). Since the Aqua and Terra satellites crossover at around 10:30 am and 1:30 pm local time respectively, the daily spatial missing patterns of Aqua and Terra AOD retrievals are different. To improve the coverage of AOD, we fitted daily linear regressions between Aqua AOD and Terra AOD (Eq. (1)-(2)). Then we predicted the missing AOD of one satellites when the AOD of another satellite exists. After the daily linear interpolation, the Aqua and Terra AODs were averaged to reflect the daily aerosol loadings.

$$AOD_{Aqua,i,g} = \mu_i + \beta_i \times AOD_{Terra,i,g} \qquad (1)$$

$$AOD_{Terra,i,g} = \mu_i + \beta_i \times AOD_{Aqua,i,g} \qquad (2)$$

Where $AOD_{Aqua,i,g}$ and $AOD_{Terra,i,g}$ represent the Aqua and Terra AOD of grid $g$ on day $i$, respectively. $\mu_i$ and $\beta_i$ represent the intercept and slope on day $i$, respectively.

**2.4 Meteorological fields and evaluation**

Various reanalysis data products, including ERA5, MERRA-2, and GEOS-FP, have been used to provide meteorological files in previous air pollution prediction models (Geng et al., 2021;Wei et al., 2021;Xiao et al., 2017). In this study, to select the best-performing meteorological data with long temporal data coverage (from 2000 to the present) and timely updating, we evaluated ERA5-Land, ERA5, and MERRA-2 meteorological datasets with meteorological measurements at regional air quality monitoring stations during 2019. The evaluation results showed that ERA5-Land data at a 0.1-degree resolution outperformed the MERRA-2 reanalysis data and the ERA5 reanalysis data (Table S1). We extracted and processed the ERA5-Land parameters, including 2-m temperature, 10-m u- and v-component of wind, surface pressure, and total precipitation, for model predictor selection. The 2-m relative humidity was calculated from the 2-m temperature and the 2-m dew point temperature.

**2.5 Construction of the time series of land use variables**

**2.5.1 Population density**

We evaluated and fused various global gridded population density data that were publicly available, including the LandScan Global Population Database from 2000 to 2019 (Dobson et al., 2000); the Gridded Population of the World (GPW) data product (version 4) for 2000, 2005, 2010, 2015, and 2020 (Doxsey-Whitfield et al., 2015); and the annual WorldPop data at a 1-km resolution from 2000-2020 (Wardrop et al., 2018;Reed et al., 2018). We linearly interpolated the GPW data for each year. For data quality evaluation, we obtained the sum population at the county or city level from 2000 to 2019 from the China City Yearbooks. The gridded population datasets were aggregated to county- or city sums and compared to the yearbook records. The LandScan data outperformed the other two datasets (Fig. S2); however, the spatial distribution of the LansScan data showed an unreasonable pattern and were excluded. As shown in Fig. S4, the LandScan data present very high population density along the road and many randomly-distributed high-population grids in certain square areas on the map. Due to the significant spatial variations in data accuracy (Bai et al., 2018;Wang et al., 2011), we fused the GPW and WorldPop data to improve data quality across space. We first fitted linear regressions between gridded population and yearbook records of each county or city that has at least 6 matched data pairs. Then, we averaged the gridded population with the $R^2$ as a weight (Eq. (3)). We selected the $R^2$ rather than the root mean square error (RMSE) as the weight because the spatial variation trends were more important than the number of populations in the prediction of $PM_{2.5}$.

$$Pop_{g(i),y} = \frac{GPW_{g(i),y} \times R^2_{GPW,i} + WorldPop_{g(i),y} \times R^2_{WorldPop,i}}{R^2_{GPW,i} + R^2_{WorldPop,i}} \qquad (3)$$

where $Pop_{i,y}$ represents the ensemble population of grid $g$ in county/city $i$ of year $y$; $GPW_{g(i),y}$ and $WorldPop_{g(i),y}$ represent the population of grid $g$ year $y$ of dataset GPW and WorldPop, respectively; and $R^2_{GPW,i}$ and $R^2_{WorldPop,i}$ represent $R^2$ of GPW and WorldPop in county/city $i$.

For counties/cities that did not have sufficient matched data pairs for regression fitting, we employed the weight of the nearest county/city for the ensemble (Fig. S3). We subsequently constrained the city level and national sum population to be consistent with the record from the China City Statistical Yearbook and the China Statistical Yearbook.

**2.5.2 Land cover**

The percentage of artificial impervious area of each 1-km modeling grid was calculated from the annual global artificial impervious area (GAIA) data at a 30-m resolution from 2000 to 2018 (http://data.ess.tsinghua.edu.cn/gaia.html). To estimate the GAIA distribution after 2018, we fitted linear regressions with data from 2016 to 2018 for each grid and extrapolated the GAIA values of 2019, 2020, and 2021. The data from 2013 to 2017 were used to evaluate the performance of this linear extrapolation. The examination results comparing the GAIA data and the first-year, second-year, and third-year extrapolated GAIA predictions showed that the $R^2$ values ranged from 0.996–0.999, 0.985–0.989, and 0.969–0.979, respectively.

**2.5.3 Road map**

Limited road maps are available in China. We collected the annual road maps from 2013 to the present from OpenStreetMap (www.openstreetmap.org) (Barrington-Leigh and Millard-Ball, 2017), a crowdsourced collaborative geographic information collection project. OpenStreetMap data have been widely used for road density analysis and the construction of world road data products (Zhang et al., 2015;Meijer et al., 2018). A previous evaluation study reported on the considerable accuracy of the OpenStreetMap data (Haklay, 2010). To evaluate the quality of the OpenStreetMap data and to fill the historical data gap before 2013, we also collected road maps of 2000, 2004, 2005, 2010, 2012, 2014, and 2015 from the survey.

We first extracted the length of various types of roads from the OpenStreetMap data and from the road survey at the grid level. We combined some road types to make the road classification more comparable in OpenStreetMap and in the survey (Table S2). To estimate the annual road distribution before 2013, we first compared the grid-level road length of 2014 extracted from OpenStreatMap and the survey map (Table S3). Then, we modify the survey map data to construct the OpenStreetMap-type gridded road length of the years that the survey map data were available by the equation listed below:

$$RL_{OSM,i,j} = \frac{RL_{OSM,i,2014} \times RL_{road,i,j}}{RL_{road,i,2014}} \qquad (4)$$

where $RL_{OSM,i,j}$ represents the OpenStreetMap-type road length of year $j$ over grid $i$ and $RL_{road,j}$ represents the survey-type road length of year $j$ over grid $i$.

After estimating the OpenStreetMap-type road length of years when the survey maps were available, we filled the gap years by weighted linear interpolation. First, we estimated the city-level sum road length of different road types by a linear mixed effects model (LME)(Meijer et al., 2018):

$$RL_{OSM,c,p,j} = (\mu + \mu') + (\beta_1 + \beta_1') \times Pop_{c,p,j} + \beta_2 \times GDP_{c,p,j} + \beta_3 \times Area_{c,p} + \beta_4 \times Ele_{c,p} + \beta_5 \times GAIA_{c,p,j} \qquad (5)$$

where $RL_{OSM,c,p,j}$ represents the sum road length of city $c$ in province $p$ of year $j$; $\mu$ represents the fixed intercept and $\mu'$ represents the province-level random intercept; $\beta_1, \beta_2, \beta_3, \beta_4,$ and $\beta_5$ represent the fixed slope of the city average population

density, per capita gross domestic product (GDP), city area, city average elevation and city average GAIA; $\beta_1'$ represents the province-level random slope of population density. The $\log_{10}$ transformation was conducted for all the continuous variables to account for the skewed distribution (Meijer et al., 2018). Stepwise linear regression was used to select the significant predictors (Meijer et al., 2018). To evaluate the LME model performance, by-year cross validation and four-year cross validation were conducted. Regarding the by-year cross-validation, we selected one-year data for model testing in sequence and used the data of the remaining years for model training. Regarding the four-year cross validation, we selected 2013, 2014, and 2015 for model testing in sequence and used the data four years after the corresponding testing year for model fitting. For example, when selecting 2013 for model testing, the data from 2017, 2018, and 2019 were used to fit the model.

After estimating the city-level sum road length, we further used the sum road length as a weight to assign the road length changes to each gap year. The equation is listed below:

$$RL_{OSM,i,j} = RL_{OSM,i,\text{start}} + (RL_{OSM,i,\text{end}} - RL_{OSM,i,\text{start}}) \times \frac{RL_{OSM,c,j} - RL_{OSM,c,\text{start}}}{RL_{OSM,c,\text{end}} - RL_{OSM,c,\text{start}}} \qquad (6)$$

where $RL_{OSM,i,\text{start}}$ and $RL_{OSM,i,\text{end}}$ represent the road length over grid $i$ of the starting and ending year with available OSM-type road data, respectively; $RL_{OSM,c,\text{start}}$ and $RL_{OSM,c,\text{end}}$ represent the road length of city $c$ of the starting and ending year with available OSM-type road data, respectively; and $RL_{OSM,c,j}$ represent the road length of city $c$ of the gap year $j$.

## 2.6 Other auxiliary datasets

We downloaded the monthly Terra MODIS NDVI (MOD13A3) at a 1-km resolution and filled the missing NDVI data by the nearest neighbour spatial smoothing approach. The average elevation data at a 30-m resolution were extracted from the Advanced Spaceborne Thermal Emission and Reflection Radiometer (ASTER) Global Digital Elevation Model (GDEM) version 2.

## 2.7 Data assimilation

All the predictors were assimilated to the 1-km MAIAC pixels by various geostatistical methods. The meteorological data and $PM_{2.5}$ predictions at a 0.1 degree resolution were downscaled to the 1-km MAIAC pixels with the inverse distance weighting method. The elevation pixels falling in each 1-km grid cell were averaged. The NDVI data were assigned to the MAIAC pixels by the nearest neighbour method.

## 2.8 Optimization and evaluation of the prediction model

To make the $PM_{2.5}$ prediction process efficient and highly accurate, we designed various examinations to optimize the model structure and identify key predictors of the $PM_{2.5}$ prediction system.

Three model structures were evaluated: model_{TwoStage} has a two-stage design with the first stage model predicting the high pollution indicator and the second stage model predicting the residual between 10-km TAP $PM_{2.5}$ predictions and measurements (Xiao et al., 2021c); model_{Resi} is a one-stage model that predicts the residual between 10-km TAP $PM_{2.5}$

predictions and measurements; and model*Base* is a one-stage model that directly predicts the PM$_{2.5}$ concentration with the 10-
km TAP PM$_{2.5}$ prediction as a predictor. Since the underestimation of high pollution events are widely reported, in addition to
the evaluations including all the test measurements, we conducted additional evaluations focusing on the prediction accuracy
of haze events when the daily average PM$_{2.5}$ concentration was higher than the 75 µg m$^{-3}$ national secondary air quality standard.
Then, we selected the critical predictors of the PM$_{2.5}$ prediction model (Fig. 1). We first constructed the full model with all the
predictors and then we removed the meteorological predictors in sequence, according to the importance of parameters
estimated from the full model. Predictors with the smallest importance were removed first. Data from 2019 were used for
model predictor optimization.
Various statistics were employed to characterize model performance. The OOB predictions were provided during the training
of the random forest that the measurements were predicted by trees that were trained with randomly selected samples without
each of these measurements. Comparing the OOB predictions with measurements in linear regression provided us with the
OOB $R^2$, the RMSE, and the mean prediction error (MPE). To evaluate the model's ability to reveal PM$_{2.5}$ variations at the
local scale and at locations without monitoring, we used the measurements from national stations for model training and the
measurements from the high-density local stations from model evaluation. These local stations are primarily located in Hebei,
Henan, Shandong, and Chengdu (Fig. S1). The evaluation results of OOB and with test data were used to optimize the model
structure and select predictors. Then, to evaluate the optimized final model's prediction ability in time, we conducted a by-
year cross validation.
Consistent with the previous TAP PM$_{2.5}$ prediction framework, the missing satellite AOD retrievals were filled by adjusting
the first layer of the decision tree and setting the availability status of AOD as the cutoff point of the first layer of the decision
trees. The performance of this gap-filling method has been fully evaluated in our previous studies (Xiao et al., 2021a;Geng et
al., 2021)
**3 Results**
**3.1 Temporal variations in predictors**
The high-resolution PM$_{2.5}$ prediction was supported by various high-resolution predictors. In addition to the 1-km resolution
MAIAC AOD, we also constructed and presented various 1-km resolution predictors, including road map, population
distribution, artificial impervious area, and NDVI (Fig. 2).
We evaluated the road length model for various road types through by-year cross validation and four-year cross validation
(Table S4). The cross-validation predictions of all road types were highly correlated with the OSM data. The four-year cross-
validation performance were comparable to the by-year cross-validation performance, indicating that the model's temporal
prediction ability was robust. The performance of the secondary road model was slightly better than that of the highway model
and primary road model, showing higher correlations between local socioeconomic factors and secondary road length relative
to highways and primary roads that are constructed nationally. The predicted highway length correctly revealed the temporal
trends of the records of highway length from the China traffic yearbooks, with correlation coefficients of 0.99. Since the road
type classifications of the OSM and China traffic yearbooks are inconsistent, we did not compare the lengths of other types of
roads. We observed a consistent increasing trend in road length for all road types across China (Fig. 2). The predicted road
maps also displayed the construction of some local landmarks, e.g., the 6th Ring Road in Beijing.
Compared to the statistical yearbook records, our ensemble population data showed $R^2$ and RMSE values of 0.74 and 0.19
million, respectively, outperforming other gridded population data (Fig. S5). The changes in population density distribution
were inconsistent across China due to the substantial internal migration during the past decades (Fig. 2). For example, we
observed that the high-speed economic development in Shenzhen city and the whole Pearl River Delta (PRD) attracted a large
migration population, but the populations in small cities in Northeast and Central China were consistent or decrease. The
artificial impervious area also increased significantly across China, especially in regions with fast economic development. The
consistent increase in NDVI over most parts of China, especially in the southeast, showed the achievement of environmental
protection in China. We found considerable missingness in MAIAC retrievals over China, especially in the southeast and
northeast regions with large populations. Thus, gap-filling is necessary to provide valuable $PM_{2.5}$ predictions across China.

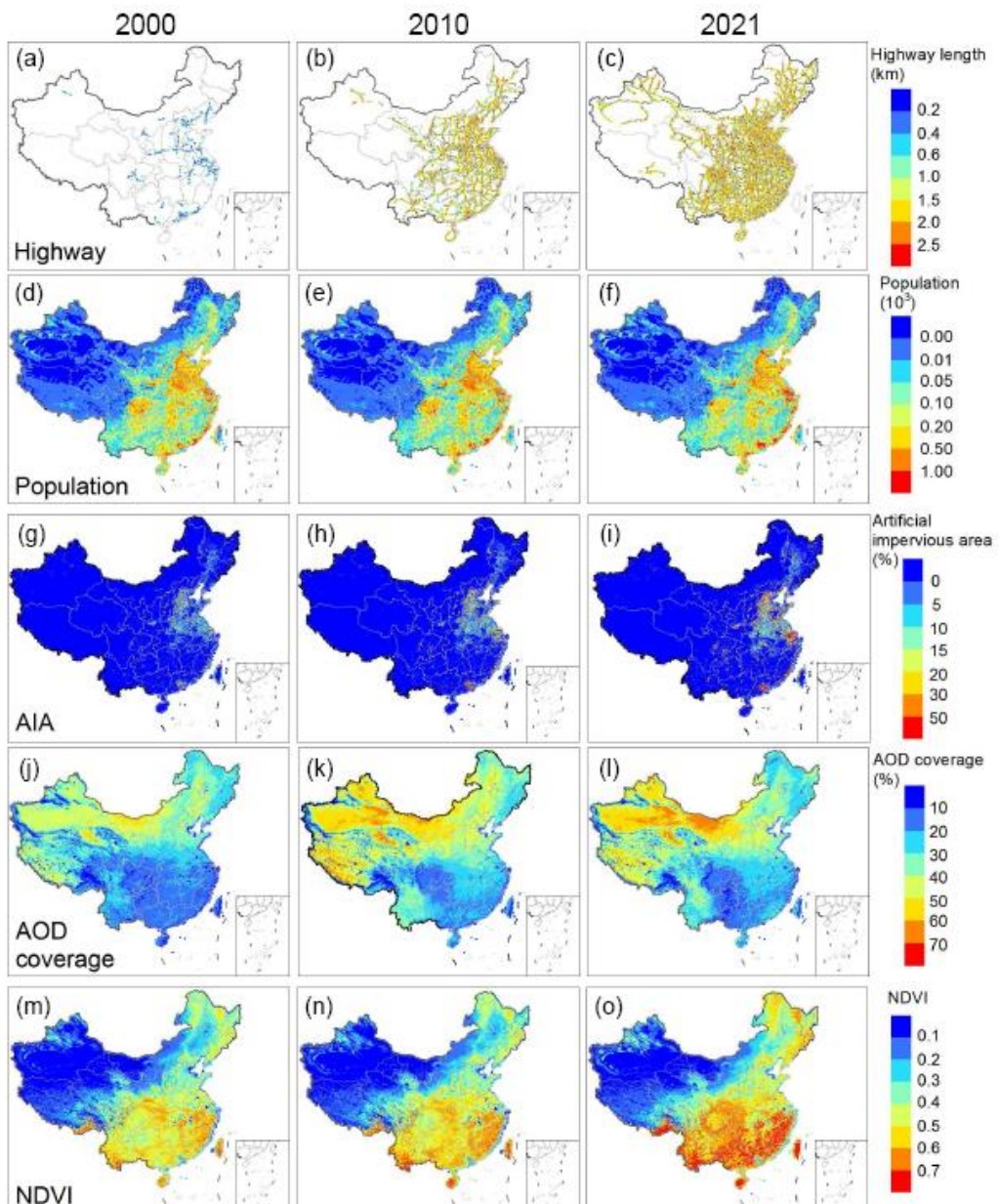


**Figure 2 Estimated annual distributions of key predictors, including highway length ((a)-(c)), population ((d)-(f)), artificial impervious area ((g)-(i)), MAIAC AOD coverage ((j)-(l)), and normalized difference vegetation index (NDVI) ((m)-(o)), in 2000, 2010, and 2021 across China.**

**3.2 Optimization of the high-resolution PM$_{2.5}$ prediction model**

Three model structures, model$_{TwoStage}$, model$_{Resi}$, and model$_{Base,}$ were examined in this study. The evaluation results showed that these candidate models performed comparably in $R^2$ in all the evaluations (Fig. 3). However, the model$_{Base}$ that directly

predicts the measurements showed significantly larger prediction error than the other two models during haze events. For some
years, e.g., 2017 and 2018, the average prediction error of the $model_{Base}$ was more than double than the prediction error of
$model_{TwoStage}$ and $model_{Resi}$. This result was consistent with our previous findings that the prediction of residuals enlarges the
response of the dependent variable to the predictors, thus benefiting the prediction of extreme events (Xiao et al., 2021c). We
did not observe significant differences between $model_{TwoStage}$ and $model_{Resi}$. Thus, considering the prediction performance and
the model fitting time expense, the $model_{Resi}$ was selected.

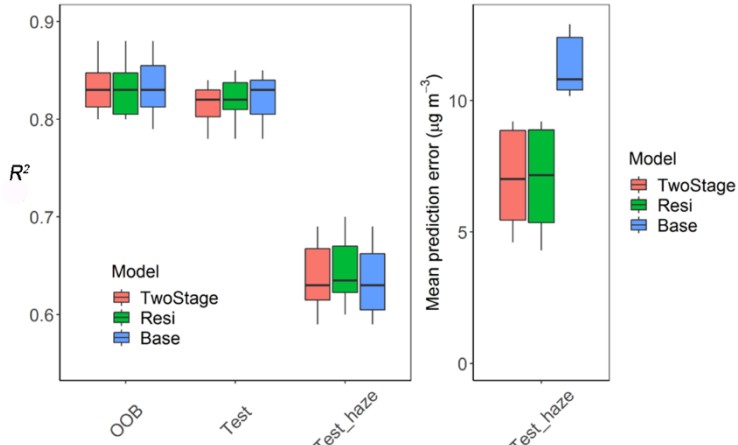


**Figure 3 The performance of $model_{TwoStage}$, $model_{Resi}$, and $model_{Base}$ in the out-of-bag evaluation (OOB), the evaluation with test data from local stations (Test) and the evaluation with test data higher than 75 µg m$^{-3}$ (Test_haze). (a) the evaluation R$^2$ (b) the average prediction error.**

We then examined the contribution of meteorological fields to the high-resolution PM$_{2.5}$ prediction (Table S5). Compared to
the full model, the OOB $R^2$ of the model without meteorological fields decreased from 0.85 to 0.80; however, the $R^2$ with test
data decreased only 0.02, from 0.85 to 0.83. This evaluation showed that the contribution of meteorological fields to high-
resolution PM$_{2.5}$ predictions was limited. Potential reasons include the coarse resolution of the meteorological data limiting
the characterization of high-resolution variations in meteorological fields or in PM$_{2.5}$ distributions. Additionally, the
meteorological effects on PM$_{2.5}$ have been considered in the 0.1-degree PM$_{2.5}$ data that served as a predictor in the model.
Comprehensively considering the model performance and the meteorological data update frequency, we removed
meteorological fields.
Table 1 summarizes the OOB performance of our final annual models and hindcast model. The model $R^2$ ranged between 0.80
and 0.84 for annual models. The small interannual variations indicated that our model was robust and provided predictions
with constant quality during the study period. The very small model mean prediction error (bias) together with the slopes close
to 1 showed the inexistence of systemic bias in the prediction models. Our model performance was comparable with previous
studies (Huang et al., 2021;Wei et al., 2019;Wei et al., 2020;Wei et al., 2021). The highcast model performed comparably in
the OOB evaluation, the test data evaluation, and the by-year cross-validation evaluation (Fig. 4), showing great accuracy and
high robustness. No considerable overfitting was observed, and no system bias was detected in the spatial prediction (test data)
and temporal prediction (by-year cross-validation) examinations.

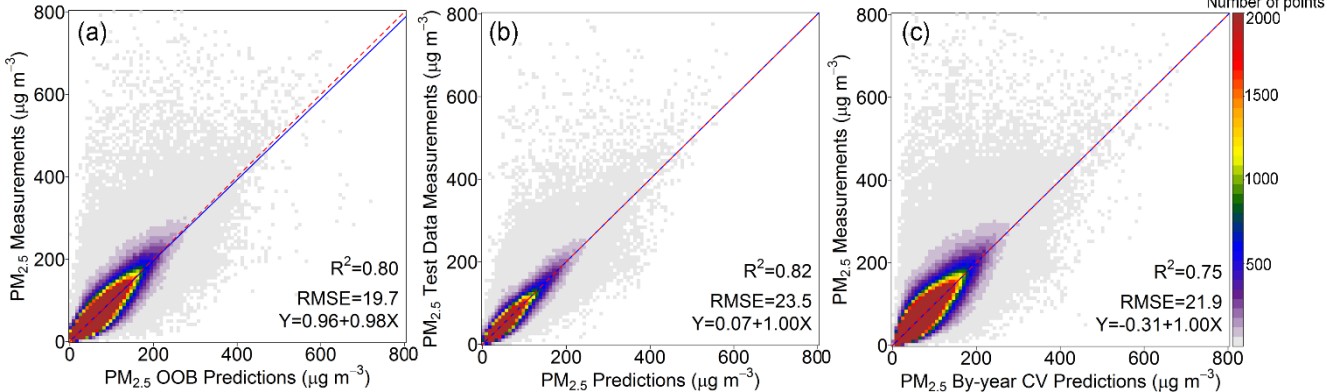


**Figure 4 Model performance of the hindcast model trained with all the data from 2013 to 2019. (a) evaluation with out-of-bag**
**predictions (b) evaluation with test data (c) evaluation with by-year cross-validation predictions**
**Table 1 Out-of-bag performance of the annual model trained with all data of each year and the hindcast model trained with all data**
**during 2013-2019**

| Model | $R^2$ | slope | RMSE ($\mu g\ m^{-3}$) | Bias ($\mu g\ m^{-3}$) |
|---|---|---|---|---|
| Annual-2015 | 0.82 | 0.99 | 20.20 | -0.06 |
| Annual-2016 | 0.83 | 1.01 | 18.24 | -0.05 |
| Annual-2017 | 0.84 | 1.00 | 16.67 | -0.02 |
| Annual-2018 | 0.82 | 0.95 | 15.94 | -0.01 |
| Annual-2019 | 0.81 | 0.95 | 16.35 | -0.04 |
| Annual-2020 | 0.80 | 0.95 | 14.96 | -0.02 |
| Hindcast | 0.80 | 0.98 | 19.6 | -0.03 |

**3.3 The spatiotemporal characteristics of the high-resolution PM$_{2.5}$ map**
The high-resolution PM$_{2.5}$ maps revealed critical local patterns of annual (Fig. 5-6) and daily (Fig. 7-8) pollution distributions
that could not be identified by the 0.1-degree resolution maps. Comparing the daily population weighted average PM$_{2.5}$
concentrations from 2000 to 2021, the number of days with PM$_{2.5}$ higher than 75 $\mu g\ m^{-3}$ were significantly reduced after 2013
across China (Fig. 5). Beijing-Tianjin-Hebei (BTH) showed high pollution levels with the haze days occurred across the year.
In recent years, benefited from the pollution control policies, high pollution days in BTH outside winter were basically removed.
The annual maps of PM$_{2.5}$ distribution in 2000 showed that the most polluted regions were located in Beijing, Hebei, and north
of Henan; in 2007 and 2013, the highly polluted regions extended and covered BTH, Shandong, Shanxi, Hunan, Sichuan basin,
and Yangtze River Delta (YRD). After 2013 with the strict pollution control policies, the air quality across China was
significantly improved and the polluted regions shrinked in 2021.

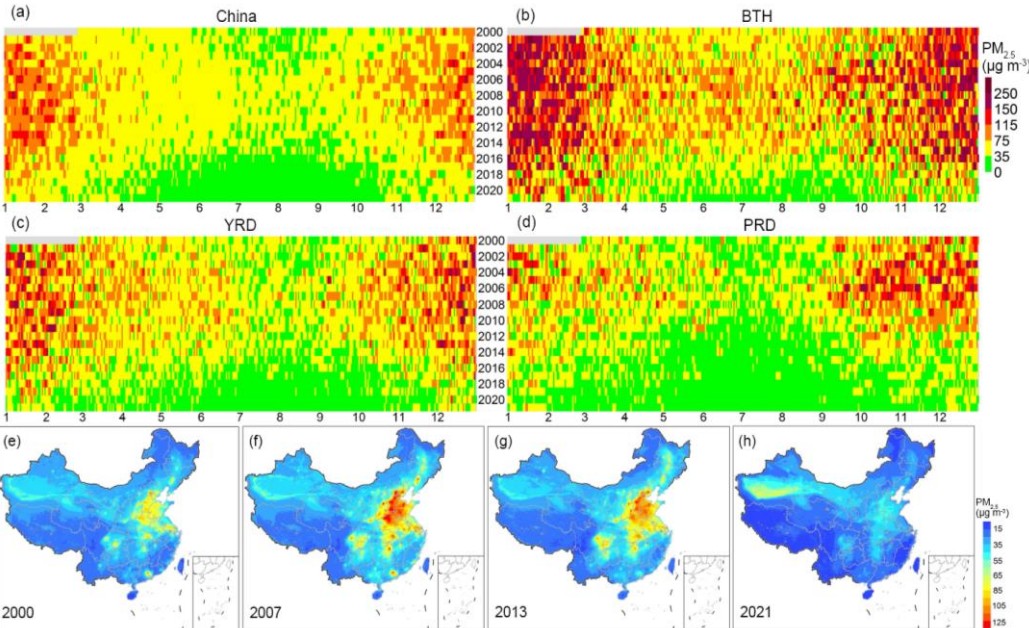


**Figure 5 Temporal variations of the daily population weighted PM$_{2.5}$ cover (a) China, (b) Beijing-Tianjin-Hebei (BTH), (c) Yangtze River Delta (YRD), and (d) Pearl River Delta (PRD) as well as the annual average PM$_{2.5}$ distribution in (e) 2000, (f) 2007, (g) 2013, (h) 2021.**

Figure 6 highlighted the variations in spatial distribution of PM$_{2.5}$ at the local scale. We compared the annual PM$_{2.5}$ anomaly,
which is the gridded PM$_{2.5}$ minus the regional average PM$_{2.5}$, in 2013 and 2021 in YRD. The pollution hotspots transferred
from city center with monitors to rural regions with limited monitoring. We found that after 2013, although the percent of days
and counties with population weighted PM$_{2.5}$ violated than the primary (35 µg m$^{-3}$) and secondary air quality standard are
continuously decreasing, the percent of days and counties with rural-county pollution higher than urban-county pollution
significantly increased. In 2013, more than half of days and counties showed higher pollution in urban counties than in rural
counties when the PM$_{2.5}$ was greater than 35 µg m$^{-3}$; however, in 2021, more than 96 % of this pollution days and counties
showed lower pollution in urban counties than in rural counties. In 2017 and 2020, all the days with PM$_{2.5}$ greater than 75 µg
m$^{-3}$ showed higher rural-county pollution than urban-county pollution. One reason of the transportation of pollution hotspot is
that the PM$_{2.5}$ reduction during 2013-2021 was much greater in city centers than in rural regions. In 2021, most regional high
pollution hotspots were transferred to around the junction of cities or towns.

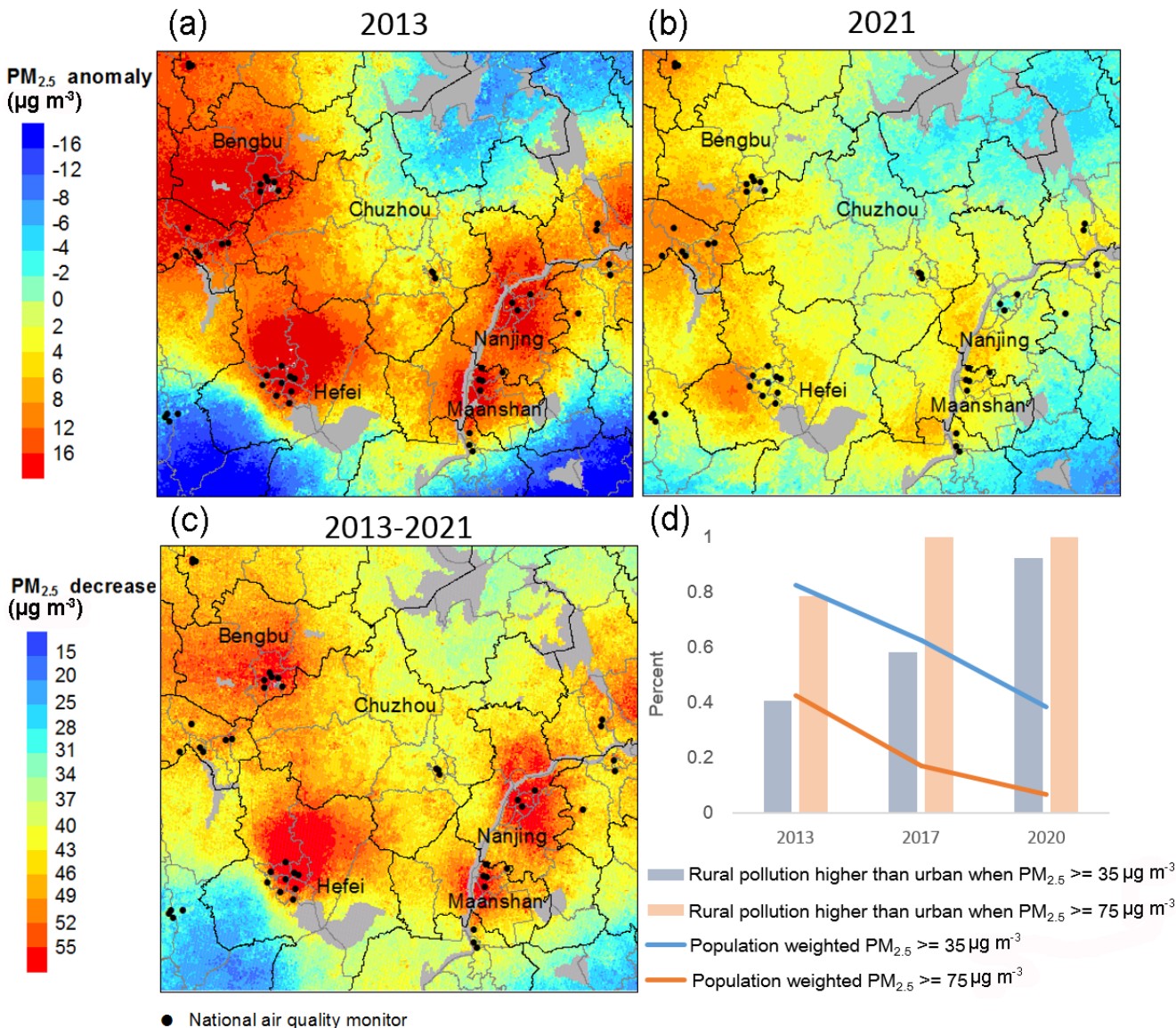

**Figure 6 The spatial distribution of annual average PM₂.₅ anomaly in (a) 2013 and (b) 2021 in YRD as well as (c) the changes in annual average PM₂.₅ between 2013 and 2021; and (d) the temporal trends in percent of days and counties with rural-county pollution higher than urban-county pollution in this region.**

The daily maps showed more short-term local pollution variations. Figure 7 shows one haze event during 18–25 November, 2013. Since 18 November, the upper cyclone moved toward northwest and the North China Plain was covered by high-pressure ridge with continuously strengthen downdraft, leading to stable atmosphere that was unfavorable for pollution control. From 18 –23 November, the pollution kept increasing and triggered the haze event. Then, since 24 November, with the upper-level ridge moved eastward to the ocean, the North China Plain was affected by the trough with increased vertical upward movement

and raised boundary layer height. Both the horizontal and vertical diffusion was improved and the pollutant concentrations
decreased sharply, leading to the end of this haze event.
The 1-km resolution pollution map was able to reveal regional characteristics. For example, the impact of local transportation
emissions was observed on some days in the populous key regions (Fig. 8). These maps also indicated the importance of
including time-varying land use data for air pollution predictions, especially in high-resolution predictions, since the land use
characteristics led to noticeable spatial variations in the pollution distribution, as expected. To examine the impact of using
temporally mismatched land use data on the retrieved spatial patterns of $PM_{2.5}$, we used historical land use data (GAIA, NDVI,
population and road map of 2000) to predict the daily $PM_{2.5}$ distribution over these key regions of the same day. The historical
land use data in 2000 led to incorrect spatial characterizations of the $PM_{2.5}$ distribution in 2015.

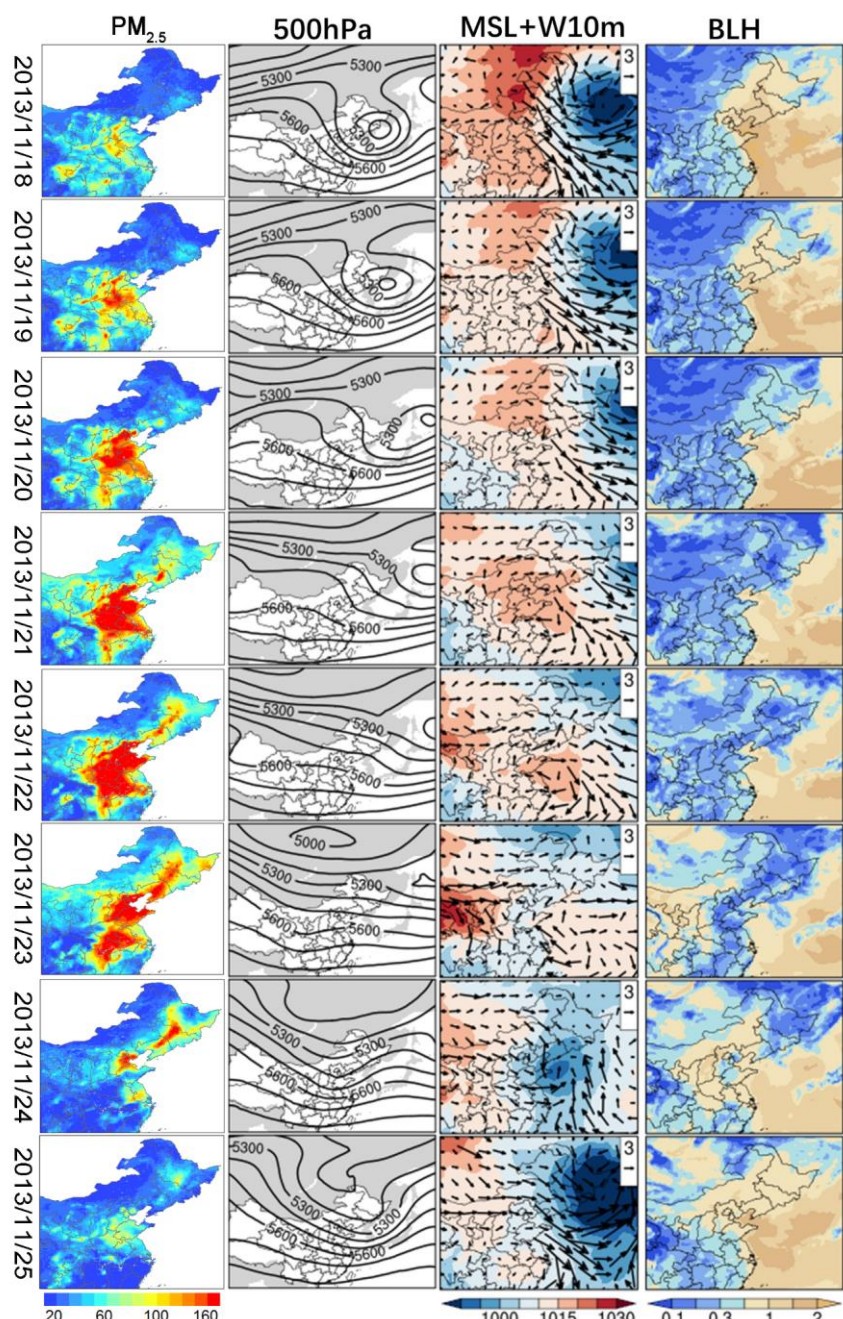

**Figure 7 Daily PM2.5 and meteorological field distributions during 18–25 November, 2013. 500hPa: the vertical height at which the**
**pressure is 500 hPa; MSL: mean sea level; W10m: wind speed and direction at 10 m; BLH: boundary layer height.**

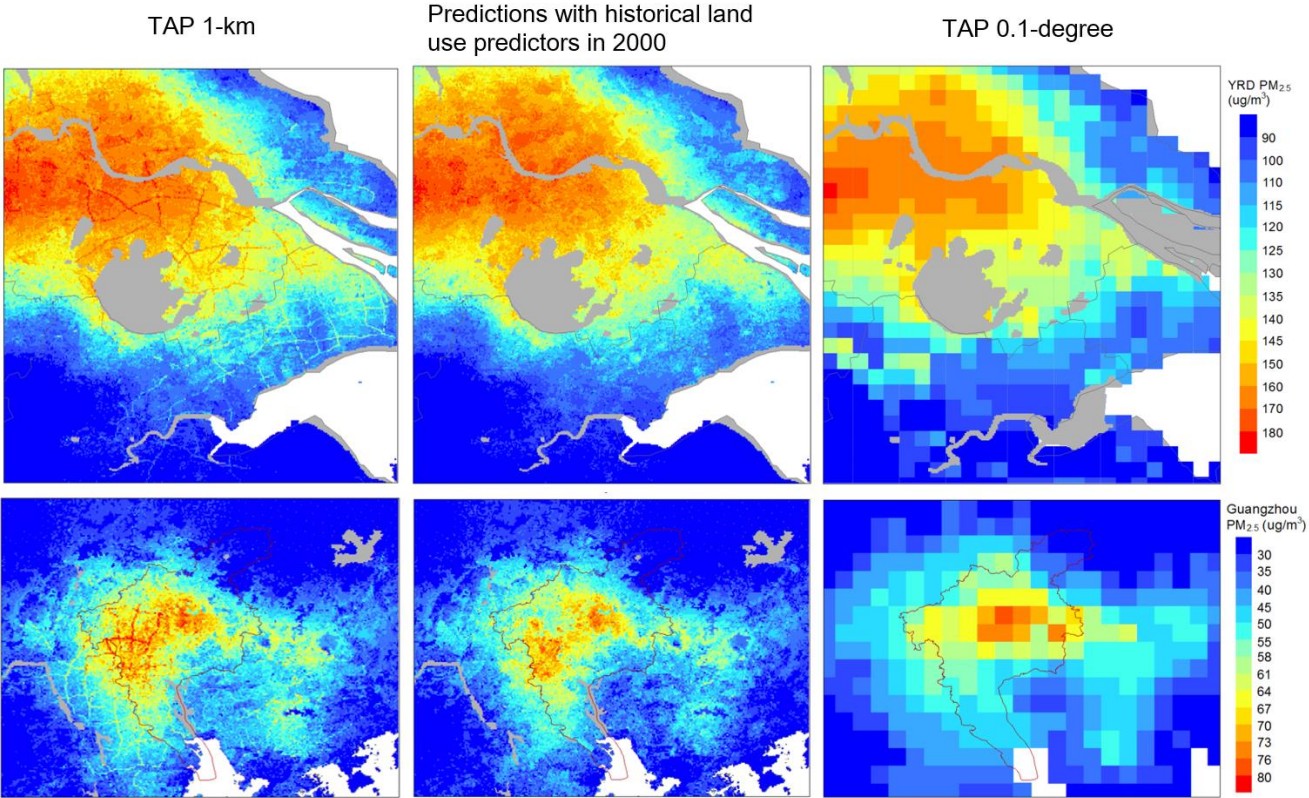

**Figure 8 The daily PM₂.₅ distribution in YRD on year 2015 day 25 ((a)-(c)) and in PRD on year 2015 day 26((d)-(f)), with the TAP 1-**
**km PM₂.₅ predictions ((a), (d)), the prediction with the historical land use predictors of year 2000 ((b), (e)), and the TAP 0.1-degree**
**PM₂.₅ predictions ((c), (f)).**
**4 DISCUSSION**
In this study, we fused the daily 10-km PM₂.₅ predictions with satellite retrievals and land use data by a random forest model
in the TAP framework and produced the open access daily average PM₂.₅ distribution at a 1-km resolution with complete
coverage from 2000 to the present. To improve the accuracy of the temporal variations in road distributions and other land use
data, we processed the annual road map by fusing the OSM data with survey data and processed the annual population
distribution by fusing the GPW and the WorldPop data. Our predictions showed an accuracy comparable with previous high-
resolution PM₂.₅ predictions, and our data were advantaged with complete coverage, time-varying land use predictors, and long
temporal coverage. Compared to previous TAP products at approximately 10-km resolution, this 1-km resolution PM₂.₅ data
product revealed more local-scale spatial characteristics of the PM₂.₅ distribution in China.
We conducted various evaluation analyses to optimize the model structure and select the appropriate predictors. When
constructing the model structure and selecting the predictors, we not only considered the prediction accuracy but also
considered the computation time, data updating frequency and long-term data availability. For example, we did not include
the POI data as predictors since we have no access to historical POI data in China, and there is no appropriate model to correctly
predict POI distributions in previous publications. Including more land use variables will certainly improve the model accuracy;
however, since historical data are unavailable, doing so will increase the uncertainty in historical predictions. Similarly,
regarding the selected spatial predictors, we constructed temporally continuous data record with various geostatistical methods
and improve the data quality by fusing data from various sources to reveal the temporal changes in land use. Including
temporally mismatched predictors for $PM_{2.5}$ prediction leads to misleading spatial patterns, especially in China with
considerable social economic development in the paste decades (Fig. 8). Additionally, we did not include any spatial and
temporal trends estimated from measurements in the hindcast model that could significantly improve the model performance
statistics in the evaluations. The measurement-based spatiotemporal trends did not necessarily reflect the pollution distribution
in regions without monitors (Bai et al., 2022a). Since the major aim of data fusing methods is to estimate the $PM_{2.5}$ variations
in regions without monitors, including measurement-based smoothing trends in space and time will hinder the achievement of
this goal. Since the predictor processing and modelling of 1-km resolution data is computationally expensive, the predictor
selection and model structure selection not only improved model robustness, but also allowed us to run a more efficient model
daily and support near real-time data updating.
Our model still has several limitations. First, although we improved the model prediction accuracy during high pollution events,
we still noticed an underestimation of $PM_{2.5}$ levels. Several reasons could explain this underestimation. The AOD retrievals
tended to misclassify high aerosol loading as cloud cover, leading to missing AOD during haze events. The CTM also hardly
predicts high pollution events. The missing satellite retrievals together with the underestimated CTM simulations resulted in
the underestimation of pollution levels. We noticed that all the predictors in the model are associated with some uncertainties
and these uncertainties together with the modelling error contributed to the uncertainties of the final $PM_{2.5}$ predictions. Thus,
the quantification of the model uncertainties and their sources could be difficult. Here we conducted various model
performance evaluations to illustrate the prediction uncertainties from different angles. We suggested the usage of the out of
bag evaluation results as an approximate of the uncertainty of $PM_{2.5}$ prediction after 2013 (Table 1), when the ground $PM_{2.5}$
monitoring is available; and the usage of the temporally cross-validation results as an approximate of the uncertainty before
2013 (Figure 4). Second, although we included some regional monitors to increase the density of monitors for model training,
the number of monitors in western China is still insufficient. Thus, the uncertainty of $PM_{2.5}$ predictions in these regions lacking
data could be larger than in the regions with dense data. However, the distribution of monitors in China basically followed the
population distribution in which populous regions hold more monitors; thus, the key regions of air pollution control have more
training data and high-quality predictions, benefiting air quality management and environmental health studies in the future.
**5. Conclusions**
In this study, we constructed a high-resolution $PM_{2.5}$ prediction model fused MAIAC satellite aerosol optical depth retrievals,
10-km TAP $PM_{2.5}$ data, and land use variables including road map, population distribution, artificial impervious area, and

vegetation index. To describe the significant temporal variations in land use characteristics resulted from the economic development in China, we constructed temporal continuous land use predictors through statistical and spatial modelling. Optimization of model structure and predictors was conducted with various performance evaluation methods to balance the model performance and computing cost. We revealed changes in local scale spatial pattern of $PM_{2.5}$ associated with pollution control measures. For example, pollution hotspots transferred from city centers to rural regions with limited air quality monitoring. We showed that the land use data affected the predicted spatial distribution of $PM_{2.5}$ and the usage of updated spatial data is beneficial. The gridded 1-km resolution $PM_{2.5}$ predictions can be openly accessed through the TAP website (http://tapdata.org.cn/).

## Data availability

The 1-km resolution $PM_{2.5}$ predictions are available on the TAP website (http://tapdata.org.cn/).

## Author contribution

QX and GG designed the study. SL ran the CMAQ simulations. LJ conducted the analysis on meteorological effects on the haze events. XM collected and provided the road maps. QX trained the $PM_{2.5}$ prediction models and conducted the spatiotemporal analyses. QX prepared the manuscript with contributions from all co-authors.

## Competing interests

The authors declare that they have no conflict of interest.

## Financial support

This work was supported by the National Natural Science Foundation of China (grant no. 42007189, 42005135, and 41921005).

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
