# Peer review of "Spatiotemporal continuous estimates of daily 1-km PM2.5 from 2000"

_EGUsphere, 2022_

## Author Response (AR1)

Reviewer 1:

This study derives a high spatial resolution (1 km) PM2.5 data from 2000 to the present over China with a range of data sets, built on the authors' previous study/framework. The study also compares several models and predictor combinations. The final open-access 1-km resolution PM2.5 product is helpful for the environment and policy-related studies in China. Generally, the manuscript is clearly written but still with room for improvement. The scope and quality of this research are suitable for publication in ACP, subject to the following concerns.

1. The authors may want to articulate what has been improved in this study from their previous TAP studies.
   **Response:** In this study, we improved the spatial resolution of $PM_{2.5}$ prediction from 10 km in previous TAP studies to 1 km, in order to describe the $PM_{2.5}$ pollution distribution at local scales. To support the $PM_{2.5}$ prediction at such a fine scale with long-term temporal coverage, we included various land use parameters and filled the data-gaps in spatial parameters to construct continuous annual spatial data. Thus, the TAP 1-km $PM_{2.5}$ predictions considered the significant changes in spatial data due to the considerable economic development in China. To summarize the improvement of this study compared to previous TAP studies, we added the following sentences in page 2:"In this study, we constructed a high-resolution $PM_{2.5}$ concentration prediction system under the TAP framework in order to provide 1-km resolution full-coverage $PM_{2.5}$ retrievals covering a long time period. To correctly reveal the spatial characteristics of $PM_{2.5}$ distribution at such a high spatial resolution, we processed MAIAC satellite retrievals as well as evaluated and constructed various temporally continuous land-use parameters with statistical and geospatial modelling that have not been included in previous TAP models." We also added the following sentence in the summary section, "Compared to previous TAP products at approximately 10-km resolution, this 1-km resolution $PM_{2.5}$ data product revealed more local-scale spatial characteristics of the $PM_{2.5}$ distribution in China."

2. It seems the authors did too much data manipulation (e.g., data/variable/model selections, different regressions, several geostatistical methods). How do such selections (and their combinations) alter the results? Please discuss.
   **Response:** Land use and other spatial parameters, e.g. road map, significantly affected the emission of air pollutants and further affected the spatial distribution of air pollution. Although these spatial parameters provides valuable information for air pollution retrieval and have been widely included in $PM_{2.5}$ perdition models, some parameters are temporally discontinuous. Including temporally mismatched predictors for $PM_{2.5}$ prediction leads to misleading spatial patterns (Fig. 8). Thus, in this study, we tried our best to the fill the data gap and improve the data quality of spatial parameters with various data sources and various geostatistical methods. As shown in Figure 8, the inclusion of annually spatial parameters improved the spatial characterization of $PM_{2.5}$.

We also conducted various model structure selection and predictor selection to create an efficient and robust model. Since the processing and modeling of 1-km resolution data is computationally expensive, although predictor selection and model structure selection may not directly improve model prediction accuracy, this process allowed us to run an efficient model daily in order to support near real-time data updating.

To clarify this, we updated the discussion section as following,"We conducted various evaluation analyses to optimize the model structure and select the appropriate predictors. When constructing the model structure and selecting the predictors, we not only considered the prediction accuracy but also considered the computation time, data updating frequency and long-term data availability. For example, we did not include the POI data as predictors since we have no access to historical POI data in China, and there is no appropriate model to correctly predict POI distributions in previous publications. Including more land use variables will certainly improve the model accuracy; however, since historical data are unavailable, doing so will increase the uncertainty in historical predictions. Similarly, regarding the selected spatial predictors, we constructed temporally continuous data record with various geostatistical methods and improve the data quality by fusing data from various sources to reveal the temporal changes in land use. Including temporally mismatched predictors for $PM_{2.5}$ prediction leads to misleading spatial patterns, especially in China with considerable social economic development in the past decades (Fig. 8). Additionally, we did not include any spatial and temporal trends estimated from measurements in the hindcast model that could significantly improve the model performance statistics in the evaluations. The measurement-based spatiotemporal trends did not necessarily reflect the pollution distribution in regions without monitors (Bai et al., 2022). Since the major aim of data fusing methods is to estimate the $PM_{2.5}$ variations in regions without monitors, including measurement-based smoothing trends in space and time will hinder the achievement of this goal. Since the predictor processing and modelling of 1-km resolution data is computationally expensive, the predictor selection and model structure selection not only improved model robustness, but also allowed us to run a more efficient model daily and support near real-time data updating."

3. How would uncertainties introduced by input data, particularly daily satellite data, impact the results. I understand that the ultimate measuring standard is to compare with observations. Nevertheless, could the authors elaborate more on the overall uncertainty of their results?

**Response:** To limit the uncertainties of satellite data, we only selected the MAIAC retrievals with the best quality according to the quality flag. Previous studies showed that the MAIC AOD showed high accuracy than the 10-km resolution MODIS AOD products compared to AERONET AOD (Mhawish et al., 2019). Another evaluation studies in China reported that more than 72% of MAIAC retrievals falling within the Expected Errors ($\pm(0.05+0.2*AOD)$))(Zhang et al., 2019). Regarding the overall quality of the prediction, since all the predictors in the model are associated with certain uncertainty and

contribute to the prediction uncertainty together with the model error, it is hard to directly simulate the uncertainty of the model. We conducted various evaluations to quantify the uncertain of our $PM_{2.5}$ prediction and we would suggest the out of bag results as an approximate of the uncertainty of $PM_{2.5}$ prediction after 2013 (Table 1), when the ground $PM_{2.5}$ monitoring is available; and the temporally cross-validation results as an approximate of the uncertainty before 2013 (Figure 4). To clarify this, we added the following sentences in the discussion section,"We noticed that all the predictors in the model are associated with some uncertainties and these uncertainties together with the modelling error contributed to the uncertainties of the final $PM_{2.5}$ predictions. Thus, the quantification of the model uncertainties and their sources could be difficult. Here we designed various model performance evaluations to illustrate the prediction uncertainties from different angles. We suggested the usage of the out of bag evaluation results as an approximate of the uncertainty of $PM_{2.5}$ prediction after 2013 (Table 1), when the ground $PM_{2.5}$ monitoring is available; and the usage of the temporally cross-validation results as an approximate of the uncertainty before 2013 (Figure 4)."

General editing issues:

1. Please use the Oxford comma to avoid ambiguity.

**Response:** We added the Oxford comma as suggested.

2. Please fix the citations in the text. For example, "Lyapustin et al., 2011b;Lyapustin et al., 2011a" in Section 2.3 is not correct, should be "Lyapustin et al., 2011a; Lyapustin et al., 2011b". "Huang et al., 2021;Wei et al., 2019;Wei et al., 2020;Wei et al., 2021" in Page 11 is surprisingly sloppy.

**Response:** The citations have been corrected as suggested.

3. Variables should be in italics.

**Response:** The variables has been changed to italics.

Monir comments:

1. Page 1, line 4, change "pollution" to "pollutants".

**Response:** The word has been changed as suggested.

2. Page 2, line 2 from the bottom, change "temporal continuous" to "temporally continuous".

**Response:** These words have been changed as suggested.

3. Page 3, some sub-plots and texts are tiny in Figure 2. I suggest the authors simplify this figure.

**Response:** Thank you for this suggestion. We showed the temporal changes in all spatial predictors in this Figure to illustrate the necessity of considering the temporal continuity of spatial predictors. To make this figure easy to read, we increased the word size of this figure.

4. Page 4, We filled …". Could you please be more specific?

**Response**: We added more details on the gap filling step as follows in page 4: "Since the Aqua and Terra satellites crossover at around 10:30 am and 1:30 pm local time respectively, the daily spatial missing patterns of Aqua and Terra AOD retrievals are different. To improve the coverage of AOD retrievals, we fitted daily linear regressions between Aqua AOD and Terra AOD. Then we predicted the missing AOD retrievals of one satellites when the AOD retrievals of another satellite are valid."

5. Page 5, line 3, what is referred to as an "unreasonable pattern"?

**Response:** We added a Figure (Supplementary Figure 4) and the following sentences in page 4 to illustrate the spatial pattern of LandScan. "As shown in Supplementary Figure 4, the LandScan data present very high population density along the road and many randomly-distributed high-population grids in certain square areas on the map."

6. Page 9, Figure 2, NDVI is missing from the caption.
**Response**: we updated the caption as "Estimated annual distributions of key predictors, including highway length, population, artificial impervious area, MAIAC AOD coverage and normalized difference vegetation index (NDVI), in 2000, 2010, and 2021 across China."

7. Page 13, Figure 6, what does a dot mean? A monitoring site?

**Response**: we added the description of the dot in the caption as "The national air quality monitors are shown as dots"

8. Page 15, Figure 7, please define those variables in the caption: "500hPa", "MLS+W10m", "BLH".

**Response**: we added the definition of these variables in the caption as following: "500hPa: the vertical height at which the pressure is 500 hPa; MSL: mean sea level; W10m: wind speed and direction at 10 m; BLH: boundary layer height"

Reviewer 2:
This manuscript presents a novel system, as part of the Tracking Air Pollution in China (TAP), that predicts the full-coverage $PM_{2.5}$ concentrations at a 1-km spatial resolution covering China. The author creatively constructed temporal continuous land-use data sets through geostatistical method and fused multisource environmental data through a machine learning method to improve prediction accuracy at a high resolution. This open-access $PM_{2.5}$ dataset with near real-time updating reveals local scale $PM_{2.5}$ variations and provides

valuable information to support studies on PM$_{2.5}$ pollution and its driven factors in China. Detailed comments are listed below:

1. The "PM2.5" in the title should be subscript.

   **Response**: The title has been updated.

2. This study used measurements from both national monitoring stations and local stations for model fitting and model evaluations. Are the data quality of measurements from different stations comparable?

   **Response**: In order to evaluate the consistency of measurements quality from national monitors and regional monitors, we matched each regional monitor with the nearest national monitor and selected such matching pairs with distance less than 0.5 degree between them. Then we compared the daily average PM$_{2.5}$ concentrations from these two types of monitors. This comparison illustrates that the measurements from regional monitors matched well with the measurements from national monitors with the linear regression R$^2$ of 0.89 (slope of 0.99 and intercept of 1.00). The average difference between these measurements are 1.6 μg/m$^3$. To make this clear, we added the following sentences in page 3, section 2.1: "In order to examine whether quality of measurements from national monitors and from regional monitors is comparable, we matched the nearest national and regional monitors and compared their measurements. Specifically, we selected such national-regional monitor pairs with distance less than 0.5 degree between them and compared their daily average PM$_{2.5}$ measurements. This comparison illustrates that measurements from regional monitors matched well with measurements from the nearest national monitors with the linear regression R$^2$ of 0.89 (slope of 0.99 and intercept of 1.00). The average difference between daily matched regional and national measurements are 1.6 μg/m$^3$."

3. In line 109, the LandScan was excluded due to an unreasonable pattern but the author did not describe or show this pattern. Please added some maps to illustrate this.

   **Response**: We added a Figure (Supplementary Figure 4) and the following sentences in page 4 to illustrate the spatial pattern of LandScan. "As shown in Supplementary Figure 4, the LandScan data present very high population density along the road and many randomly-distributed high-population grids in certain square areas on the map."

References:
Bai, K., Li, K., Guo, J., and Chang, N.-B.: Multiscale and multisource data fusion for full-coverage PM2.5 concentration mapping: Can spatial pattern recognition come with modeling accuracy?, ISPRS Journal of Photogrammetry and Remote Sensing, 184, 31-44, https://doi.org/10.1016/j.isprsjprs.2021.12.002, 2022.
Mhawish, A., Banerjee, T., Sorek-Hamer, M., Lyapustin, A., Broday, D. M., and Chatfield, R.: Comparison and evaluation of MODIS Multi-angle Implementation of Atmospheric Correction (MAIAC) aerosol product over South Asia, Remote Sensing of Environment, 224, 12-28,

https://doi.org/10.1016/j.rse.2019.01.033, 2019.

Zhang, Z., Wu, W., Fan, M., Wei, J., Tan, Y., and Wang, Q.: Evaluation of MAIAC aerosol retrievals over China, Atmospheric Environment, 202, 8-16, https://doi.org/10.1016/j.atmosenv.2019.01.013, 2019.

---

## Author Response (AR2)

The editor's comments:

Please follow the ACP guidelines carefully (https://www.atmospheric-chemistry-and-physics.net/submission.html) to correct the mathematical notation and terminology, English, Figures & tables, and reference. For example, units in Table 1 are wrong and also in a wrong format in the main text. R^2 should be defined at the first occurrence and be italic. The equation format looks terrible and doesn't match with the text format of variables or non-variables. The linear regression method should be clarified. It is better to name figures in SI differently (e.g., Fig. SX instead of Fig. X to avoid confusion). The reference format also has problems (e.g., title capitalization and subscript, doi and journal format, etc.).

**Response:** Thank you for your comments. We have updated all the figures following the ACP guideline. We corrected the units and the format of "$R^2$" in Table 1, in the main text, and in figures. We added the definition of $R^2$ and corrected the locations of definition on other abbreviations. We adjusted the format of equations and added equations of the gap-filling linear regressions. We changed the figure names in SI as suggested and corrected the reference format. We noticed that Word compressed some figures leading to poor figure quality, thus we attached the figures in their original formats.